# RADLADS: Rapid Attention Distillation to Linear Attention Decoders at Scale

**Daniel Goldstein**
Recursal AI
Eleuther AI
dan@recursal.ai

**Eric Alcaide**
USI, IDSIA
Eleuther AI
eric.alcaide@usi.ch

**Janna Lu**
Recursal AI
GMU
janna@recursal.ai

**Eugene Cheah**
Recursal AI
Eleuther AI
eugene@recursal.ai

## Abstract

We present Rapid Attention Distillation to Linear Attention Decoders at Scale (RADLADS), a protocol for rapidly converting softmax attention transformers into linear attention decoder models, along with two new RWKV-variant architectures, and models converted from popular Qwen2.5 open source models in 7B, 32B, and 72B sizes. Our conversion process requires only 350-700M tokens, less than 0.005% of the token count used to train the original teacher models. Converting to our 72B linear attention model costs less than $2,000 USD at today's prices, yet quality at inference remains close to the original transformer. These models achieve state-of-the-art downstream performance across a set of standard benchmarks for linear attention models of their size. We release all our code [here] and models [here] under the Apache 2.0 license, with the exception of our 72B models which are also governed by the Qwen License Agreement.

## 1 Introduction

Linear attention transformer variants employing compressive state have begun to match or even exceed traditional softmax attention based transformers in quality across many metrics. (Peng et al., 2025) This is important for long sequences at inference time, as linear attention is computable in O(1) time per token instead of O(N) time for softmax transformers, and avoids the expensive memory bandwidth usage of a Key Value cache (Arora et al., 2023). However, the cost of training usable large models at scale is prohibitively costly for all but the largest organizations. This is especially true for large language models, where SoTA results have often required training on over ten trillion tokens of data. (Qwen et al., 2025) (Grattafiori et al., 2024)

| Name | Tokens required | Stages | Relative score[a] (%) | | |
| --- | --- | --- | --- | --- | --- |
| | | | Lambada↑ | MMLU↑ | Others↑ |
| SUPRA (Mercat et al., 2024) | 100B | 1 | 91.3 | 21.6 | 86.9 |
| LoLCats/Hedgehog[b] (Zhang et al., 2024a) | 40M | 2 | - | -28.8 | 63.5 |
| MOHAWK (Bick et al., 2024b) | 3-5B | 3 | 93.8 | -4.7 | 92.4 |
| Mamba in the Llama (Wang et al., 2024) | 20B | 2 | 60.8 | 51.9 | 75.7 |
| DiJiang[b] (Chen et al., 2024) | 40B | 1 | - | 72.0 | - |
| ARWKV (Yueyu et al., 2025) | 60M/830M | 2/3 | 89.3 | 80.1 | 93.5 |
| Llamba (Bick et al., 2025) | 8-12B | 3 | [c]95.1 | [c]83.7 | [c]98.7 |
| **RADLADS (ours)** | **350-700M** | **3** | **98.3** | **92.4** | **101.3** |

Table 1: Recent softmax attention to purely recurrent model conversions up to 8B parameters
[a] Relative score is computed as: $\frac{s-r}{t-r}$ where $s$ is student accuracy, $t$ is teacher accuracy, and $r$ is chance of a correct random guess. See Table 9 for absolute scores. 0-shot except for 5-shot LolCATs mmlu. Lambada_openai (Paperno et al., 2016) and MMLU (Hendrycks et al., 2021a) shown. Others means avg. of arc_c_norm, arc_e (Clark et al., 2018), piqa (Bisk et al., 2020), winogrande (Sakaguchi et al., 2021), hellaswag_norm (Zellers et al., 2019).
[b] Neither model weights nor the missing scores were published for this pure RNN model by its authors.

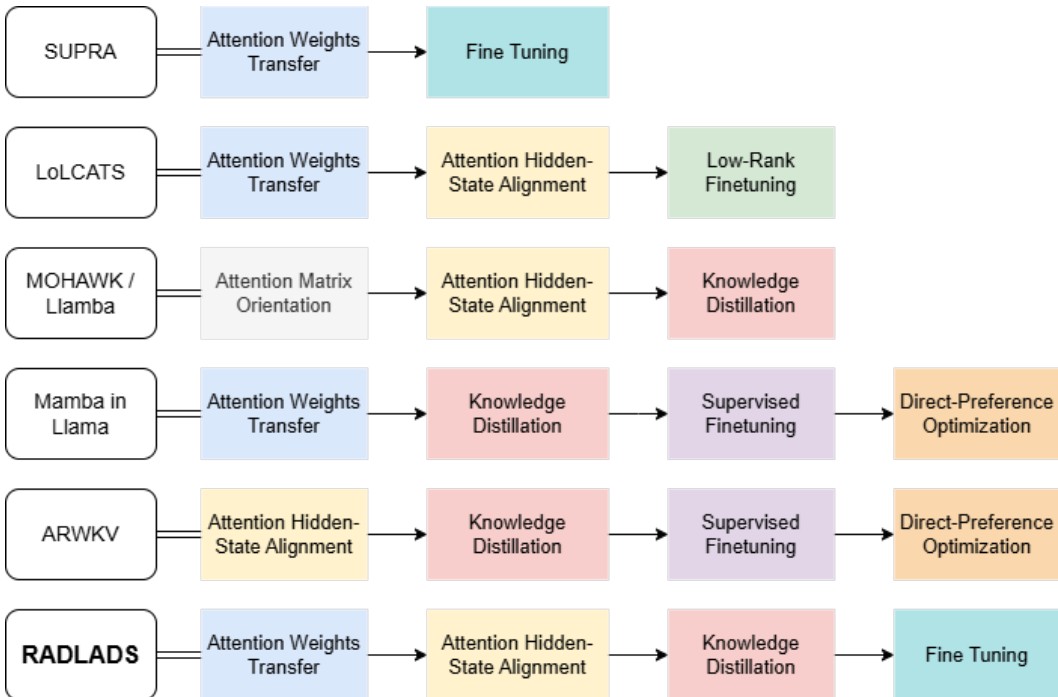

Figure 1: Conversion Method Steps

We present a distillation method that yields comparable performance to the original teacher model, while using only between 350-700 million tokens, less than 0.005% of the tokens required to pre-train SoTA transformers. This approach enables the creation of large linear attention language models without incurring the extreme costs of multi-trillion token training runs. It also opens up new avenues for researchers who work on the next generation of compressive state attention variants to test, train, and release models containing their new designs at scale.

While converting models, we found that pre-existing RWKV architectural choices were sometimes imperfectly matched to the needs of conversion. As a result, we developed two new architectures RAD-RWKV6 ("RADFinch") and RAD-RWKV7 ("RADGoose") based on RWKV-6 and RWKV-7 that allow conversion to proceed smoothly and the resultant model to be faster to inference than had we used the original RWKV-6 and RWKV-7 designs. We share these new architectural details, as well as some of the reasons behind the changes. We hope that other researchers may find this useful in converting transformers to their own linear attention designs.

Our main contributions are:

- Details of the RADLADS distillation recipe, including stepwise instructions, specific hyperparameters, token counts, and dataset choice.
- Two new simplified RWKV based architectures: RAD-RWKV6 ("RADFinch") and RAD-RWKV7 ("RADGoose"), used to convert softmax attention efficiently to linear attention.
- Public release of several converted Qwen models: QRWKV6-7B-Base, QRWKV6-7B-Instruct, QRWKV7-7B-Instruct, QRWKV6-32B-Instruct, QRWKV6-72B-Instruct.
- Open source conversion code, allowing anyone to adapt the RADLADS process to a transformer of their choice and emit a highly performant converted model, including for other recurrent linear attention and/or compressive state architectures.

## 2 Background

There have been many attempts at efficiently converting pre-trained transformer models to pure linear attention or other recurrent compressive state architectures. Early work (Gerstenberger et al., 2020) performed full model logit distillation to a freshly initialized student model, and

required a very long training cycle. T2R (Kasai et al., 2021) attempts to reduce this training time by keeping most of the original model intact. It swaps softmax attention for traditional linear attention with a learnable MLP feature map, and finetunes the resultant model on approximately 3.5 billion tokens of data. Other projects like DiJiang (Chen et al., 2024), XATL (Choi, 2024), and SUPRA (Mercat et al., 2024) combine these two techniques, but still require 100B tokens of training or more, and exhibit lacking performance on some popular benchmarks like MMLU.

More recently, new model architectures such as GSA (Zhang et al., 2024b) have been proposed, which are designed to adapt well to a SUPRA-like conversion process. Post-conversion downstream performance is improved, but these still require long training cycles.

Other works such as Mamba in the Llama (Wang et al., 2024) have focused on a pipeline of progressive distillation followed by supervised fine-tuning (Kim & Rush, 2016) and direct preference optimization (Rafailov et al., 2024). Mamba in the Llama uses 20B tokens of training. However, it focuses on hybrid models rather than pure RNNs, and performs poorly when the softmax attention is removed completely.

A recent development in the conversion process, featured in LoLCats (Zhang et al., 2024a) and MOHAWK (Bick et al., 2024a) has been to separate the conversion process into two phases: attention alignment and full model knowledge distillation. The first phase aligns attention scores and/or attention hidden states with the teacher model, and the second performs traditional logit distillation. MOHAWK still requires training on a large amount of data. LoLCats (Hedgehog version) requires only 40m tokens of training, but benchmark scores remain low, with MMLU accuracy lower than random guessing. To address this deficiency, they create a hybrid with full softmax attention in a sliding window (SWA), which improves scores but is no longer a purely recurrent model. We theorize that their low scores are due to the use of LoRAs for training, as well as vanilla linear attention rather than a more advanced recurrent architecture. This is most evident when examining their relative MMLU scores, which even with the inclusion of SWA remain low compared to ours.

We shared our early code, techniques, and 32B model with another team, who worked with those and subsequently published ARWKV (Yueyu et al., 2025), which converts Qwen2.5-7B-Instruct to use a standard RWKV-7 sequence mixer. As we discuss later, specific choices for weight transfer, hyperparameters, dataset, and architecture matter significantly. This is visible in the significantly higher downstream performance of our own Qwen2.5-7B-Instruct conversion, as shown in Tables 1 and 4.

## 3   Design choices

Our work builds upon this varied foundation of historical attempts to convert softmax attention transformers to linear attention, and we attempt to use all of the best parts in a single process. We transfer attention weights, align attention hidden states, distill, finetune, develop a strong and appropriate RNN attention-replacement architecture, use a good dataset, and find the proper sequence of distillation hyperparameters. Combining all of these together results in lower training token counts while achieving a new state-of-the-art in pure RNN downstream performance. The reduction in token count allows us to affordably convert very large models, up to 72 billion parameters so far.

We find that choosing a good RNN architecture is an important part of any highly successful conversion process. T2R, SUPRA, and LoLCats all use traditional linear attention or make small changes (e.g. removing the denominator in favor of normalization). Other modern RNN architectures have been shown to be more expressive than linear attention, even when including such changes. (Sun et al., 2023; Yang et al., 2024; Peng et al., 2025) But choosing a strong architecture is not always enough. We initially used an unmodified RWKV-6 architecture in our experiments, and found disappointing performance. Removing the off-by-one decay and bonus by using a Gated Linear Attention kernel allowed the model to fit the original softmax attention hidden states much more closely during step 1. And we find that RWKV-7 fits even closer and more rapidly during this step, resulting in much lower distillation loss with even less compute.

Some components of modern RNN architectures that appear important for pretraining turned out to have little to no effect during conversion. And some have useful effects in one architecture but

not others. One interesting example is RWKV tokenshift, a variety of 1D short convolution. It was useful in RAD-RWKV6, but conferred essentially no benefit in RAD-RWKV7. Another such case is gating, which was extremely beneficial only at full rank in RAD-RWKV6, but performed perfectly well in RAD-RWKV7 with reduced rank. Careful testing of each component can help discover the most efficient variant for a given architecture for use in softmax attention conversion.

Choice of dataset appears to matter quite a bit. After trying several custom datasets, as well as fineweb and fineweb-edu (Lozhkov et al., 2024), we settled on DCLM (Li et al., 2024b) for all our conversions. We theorize that the best choice of dataset may depend upon the teacher model's pre-training data distribution. DCLM worked exceptionally well for us when converting Qwen models.

Training for the correct number of tokens and with good choices for learning rates was also very important, both for efficiency of training and for final downstream performance. More training was not always better. We arrived at our final hyperparameters after theorizing that most knowledge resides in the teacher model's MLPs and embeddings. Therefore, we start at a high learning rate to rapidly align the attention hidden states, annealing to approximately the final learning rate that the teacher model saw during pretraining. We tried various learning schedules, and cosine worked best. We keep the learning rate for the MLPs fixed during steps 2 and 3. This is an attempt to avoid catastrophic forgetting or overly significantly changing the knowledge encoded in the original teacher MLPs.

## 4 Method

In RADLADS, our student model architecture is an exact copy of the teacher model architecture, with the exception of the attention blocks. These are replaced with a choice of purely recurrent attention-replacement sequence mixing blocks. In this paper we detail two such replacements, which are based on RWKV6-C2 and RWKV7, which we name RAD-RWKV6 ("RADFinch") and RAD-RWKV7("RADGoose") respectively. Our student models retain the runtime repetition of keys and values from Grouped Query Attention (GQA) (Ainslie et al., 2023) in the teacher model, if present. Depending on the attention-replacement chosen, we optionally include rotary positional embeddings (RoPE) (Su et al., 2023) from the teacher model. In cases where we include RoPE, it is applied at exactly the same point as in the teacher model.

See Appendix A for detailed formulas for RAD-RWKV6 and RAD-RWKV7.

The protocol for converting a multilayer transformer model from one sequence mixing operation to another can be described as 3-step process:

- **Setup**: Attention Weights Transfer. All attention related weights ($W_q, W_k, W_v, W_o$) from the teacher model are transferred to the student model.
- **Step 1**: Attention Hidden State Alignment. Each student sequence mixing layer is trained to approximate the corresponding teacher model attention layer's hidden state outputs. This can be done sequentially or in parallel.
- **Step 2**: Knowledge Distillation. All model layers are trained together to approximate original model de-embedding output logits via Kullback-Leibler Divergence Loss.
- **Step 3**: Fine Tuning. The new model is fine-tuned at longer contexts.

### 4.1 Setup: Attention weights transfer

When equivalent parameters are available (e.g. query = receptance, key = key, value = value, output = output) we initialize them to the teacher weights, and otherwise to a standard pretraining initialization (e.g. $w$ in RWKV6 or RWKV7 models). Some weights, such as tokenshift, are set such that they mimic the teacher model and have no immediate effect. The goal is to initially match the teacher but learn new more expressive behavior as needed throughout distillation.

### 4.2 Step 1: Attention hidden state alignment

Following notation from MLPmixer (Tolstikhin et al., 2021), we divide the layers of traditional transformers into pointwise operations (MLPs) and sequence mixing operations (Attention layers

in standard transformers). In Step 1, we hold a frozen copy of the full teacher model in memory, and add a second trainable attention-replacement layer in parallel alongside each teacher attention layer, created with the replacement sequence mixing operation of choice (ex. RAD-RWKV6 or RAD-RWKV7). For loss, we employ an L2 distance (or optionally, mean squared error) objective between the student and teacher sequence-mixer hidden state outputs.

As noted earlier, this step can be done sequentially layer by layer, or in parallel with all layers at once. In practice, we train all layers at once using a single optimizer and whole model forward/backward pass.

The sequence length of this step is usually 512. We find that 100M tokens is enough to converge to a low stable loss during step 1. We run this step with a cosine annealed learning rate beginning at 1e-3 and ending at 1e-5 or 7e-6. The final learning rate should be similar to the final learning rate the teacher saw during pretraining.

At the end of this step we remove the teacher attention layers from the model, leaving only a complete student model.

### 4.3 Step 2: Knowledge distillation (modelwise approximation of logits)

During this step, we load a separate copy of the full teacher model and train the student model to approximate the logits of the teacher model in a similar fashion to traditional teacher-student knowledge distillation training. (Hinton et al., 2015) We use loss equal to the Kullback-Leibler divergence of the student and teacher logits.

Empirically, we find that the model resulting from this step reproduces most of the initial model performance across a variety of benchmarks, but suffers from lack of context usage.

The sequence length of this step in our original protocol was 512. We ran this step with a flat learning rate of 1e-5 or 7e-6. This should be similar to the final learning rate the teacher saw during pretraining.

### 4.4 Step 2a: Knowledge distillation with context-length extension

More recently, we found that using a longer sequence length such as 4096 during step 2 can largely avoid the need for step 3. This step 2a is an optional replacement for steps 2 and 3. A naive run with increased length would take much longer and require much more data. But by supporting separate learning rates for the sequence mixer versus the rest of the model, we were able to adjust hyperparameters to reduce batch size and maintain the same total distillation token count and approximate speed. We find that keeping the sequence mixer learning rate high (1e-4) during most of step 2a and annealing it rapidly near the end of training and then continuing at a low rate (1e-5 or 7e-6), the model is able to adapt to the longer context length during distillation.

### 4.5 Step 3: Context length extension

During this step we train the model on longer sequences with the objective of enhancing its long-context modeling capabilities. We use regular cross-entropy loss on target labels with no teacher model. This process is done as a separate step for efficiency and memory reasons, instead of simply increasing the context length during steps 1 & 2. Steps 1 & 2 require part or all of both teacher and student models to be held in memory at the same time, while this is no longer the case in step 3. Step 3 also runs faster since it has only one model to execute, making it a better candidate for longer context lengths. It is possible to reduce the memory usage in step 2 by saving teacher logits to disk or having them generated by a separate machine, but the amount of VRAM saved would be relatively modest, since the teacher model requires neither optimizer states nor backpropagation.

We run this step with the same flat learning rate of 1e-5 or 7e-6 as step 2.

We provide an alternative method for step 3 in situations where VRAM is at a premium. In this alternative step 3a, we freeze all weights except for decay and tokenshift (if present), and increase the learning rate to $1 \times 10^{-4}$.

## 5    Training details and hyper-parameters

We use the DCLM (Li et al., 2024b) dataset for all three steps in the RADLADS process, with the following hyper-parameters and token counts. For more optimizer details see Appendix B.

| Step | Tokens | $LR_{main}$ | $LR_{att}$ | $LR_{final}$ | $LR_{decay}$ | seqlen | batch size | adam betas, eps |
|---|---|---|---|---|---|---|---|---|
| 1 | 100M | $1 \times 10^{-3}$ | $1 \times 10^{-3}$ | $1 \times 10^{-5}$ | cosine | 512 | 32 | 0.9, 0.95, $1 \times 10^{-8}$ |
| 2 | 250M-700M | $1 \times 10^{-5}$ | $1 \times 10^{-5}$ | $1 \times 10^{-5}$ | none | 512 | 96 | 0.9, 0.95, $1 \times 10^{-8}$ |
| 2a | 600M | $1 \times 10^{-5}$ | $1 \times 10^{-4}$ | $1 \times 10^{-5}$ | s-cos-s [a] | 4096 | 32 | 0.9, 0.95, $1 \times 10^{-8}$ |
| 3 | 100M | $1 \times 10^{-5}$ | $1 \times 10^{-5}$ | $1 \times 10^{-5}$ | none | 16384 | 96 | 0.9, 0.95, $1 \times 10^{-8}$ |

Table 2: Hyper-parameters for each training step

$LR_{main}$ is the learning rate applied to most of the model, while $LR_{att}$ is the learning rate applied to the replacement sequence mixer. [a] In step 2a, cosine decay is employed between 83.3% and 87.5% of the tokens trained, with the learning rate remaining stable outside of that region.

| Model Size | GPU | Step 1 Tokens | Opt | Hours | Step 2 Tokens | Opt | Hours | Step 3 Tokens | Opt | Hours |
|---|---|---|---|---|---|---|---|---|---|---|
| 7B | 8x Mi300X | 100M | DS1 | 0.75 | 500M | DS1 | 5.5 | 100M | DS1 | 1 |
| 32B | 8x Mi300X | 100M | DS1 | 2.5 | 500M | FSDP | 27.0 | 100M | FSDP | 3 |
| 72B | 8x Mi300X | 100M | FSDP | 7.50 | 500M | FSDP | 54.0 | 100M | FSDP | [a]6 |

Table 3: Approximate conversion timing guidance
[a] 16384 ctxlen does not fit on a single node at 72B scale

## 6    Language modeling performance

In Table 4 we compare pure RNN models output from various conversion methods across a set of standard benchmarks. These converted models were trained from a variety of different teacher models and vary in terms of parameter count. Therefore, we compare them via relative score: the ratio of benchmark accuracy scores between each student and teacher model, first subtracting the random chance accuracy for each benchmark. Negative ratios indicate that a model performed worse than random chance. Note that on nearly every benchmark, the RADLADS "QRWKV" models obtain higher score ratios than models converted via any other method. ("QRWKV" is our shorthand for "Qwen with RWKV attention") In Table 5 QRWKV6-72B-Instruct demonstrates a new state-of-the-art in downstream performance for a pure RNN language model.

Next, in Tables 6 and 7 we show models converted to hybrid-attention variants, each containing some amount of softmax attention. Even when compared to these hybrid models that still employ softmax attention, the pure RNN QRWKV models show the highest MMLU relative scores, and achieve similar or better relative scores on other benchmarks.

Shown in Tables 10 and 11, we also tested a subset of models on harder and long context tasks gsm8k(Cobbe et al., 2021), HumanEval(Chen et al., 2021), passkey, and RULER(Hsieh et al., 2024). These demonstrate some of the limitations of distilling transformers to RNNs, especially for complex long sequence tasks.

To test reasoning performance, we distilled a RAD-RWKV7 model from the reasoning model Qwen3-8B. In Table 12, we show the Minerva Math (Hendrycks et al., 2021b; Lewkowycz et al., 2022) benchmark, a superset of the popular MATH500 (Lightman et al., 2023) test. Of interest is the fact that the converted model does not appear to improve as much with additional output tokens as the teacher model. This may indicate a need for improved context usage in the converted model.

Using DCLM we saw repetitive looping behaviors during reasoning, but found these can be eliminated by training on a mixture of 90% DCLM and 10% OpenThoughts (Guha et al., 2025), leading to a reasonable chain of thought. It is possible that reasoning datasets must be much more aligned with the teacher model than the ones we are using.

| Name | lmbda↑ | mmlu↑ | arc_c↑ | arc_e↑ | hella↑ | piqa↑ | winog↑ |
|---|---|---|---|---|---|---|---|
| Mistral-7B-v0.1-SUPRA | 91.3 | 21.6 | 72.2 | 91.1 | 93.0 | 93.7 | 84.6 |
| Mamba2-Llama3.0-8B-Instruct | 60.8 | 51.9 | 72.4 | 86.8 | 90.1 | 90.3 | 39.1 |
| MOHAWK-Phi1.5-1.3B | 93.8 | -4.7 | 83.0 | 96.8 | 93.6 | 95.9 | 92.7 |
| DiJiang-Llama2.0-7B[a] | - | 72.0 | [c]97.3 | [c]73.2 | 90.6 | 97.5 | - |
| Llamba-8B[b] | 95.1 | 83.7 | 98.3 | 101.4 | 96.9 | 99.4 | 97.5 |
| LoLCatsHedgehog-Llama3.0-8B[a] | - | -2.9 | 53.4 | 86.4 | 75.0 | 88.6 | 14.3 |
| ARWKV-7B | 89.3 | 80.1 | 90.6 | 96.9 | 93.4 | 99.3 | 87.4 |
| *Our RADLADS models:* | | | | | | | |
| **QLinAtt-7B-Instruct** | 90.7 | 64.0 | 94.6 | 96.7 | 90.7 | 98.0 | 90.2 |
| **QRWKV6-7B-Instruct** | 97.0 | 87.1 | 104.3 | 99.8 | 97.4 | **1.030** | 1.007 |
| **QRWKV6-7B-Instruct-RoPE** | 96.9 | 88.0 | 105.4 | 100.3 | 97.1 | 101.8 | 98.1 |
| **QRWKV7-7B-Instruct-RoPE** | 98.2 | **92.4** | 102.6 | 100.4 | **97.9** | 101.3 | 104.1 |
| **QRWKV7-7B-Instruct-from72B[b]** | **101.6** | 89.3 | **107.1** | **103.3** | 95.9 | **102.6** | 113.9 |
| **QRWKV6-32B-Instruct** | 98.6 | 90.9 | **106.6** | **103.7** | 96.4 | 100.5 | **123.5** |
| **QRWKV6-72B-Instruct** | 1.004 | 89.9 | 101.5 | 100.8 | **97.4** | 96.8 | 112.3 |

Table 4: Relative scores (%) for student RNN models

Relative score $= \frac{s-r}{t-r}$, with student accuracy $s$, teacher accuracy $t$, and chance of correct random guess $r$.

| Name | lmbda↑ | mmlu↑ | arc_c↑ | arc_e↑ | hella↑ | piqa↑ | winog↑ |
|---|---|---|---|---|---|---|---|
| Mistral-7B-v0.1-SUPRA | 69.2 | 33.1 | 45.8 | 75.9 | 77.1 | 80.1 | 70.3 |
| teacher: Mistral-7B-v0.1 | 75.8 | 62.4 | 53.8 | 80.9 | 81.0 | 82.1 | 74.0 |
| Mamba2-Llama3.0-8B-Instruct | 43.9 | 45.2 | 48.0 | 74.1 | 70.8 | 75.8 | 58.6 |
| teacher: Llama3.0-8B-Instruct | 72.2 | 63.9 | 56.7 | 81.6 | 75.8 | 78.6 | 71.9 |
| MOHAWK-Phi1.5-1.3B | 50.1 | 24.2 | 44.1 | 74.0 | 60.2 | 75.5 | 71.7 |
| teacher: Phi1.5-1.3B-base | 53.4 | 41.8 | 48.0 | 75.6 | 62.6 | 76.6 | 73.4 |
| DiJiang-Llama2.0-7B[b] | - | 40.7 | 42.7 | 62.6 | 69.4 | 77.5 | - |
| teacher: Llama2.0-7B | - | 46.8 | [c]46.3 | [c]76.4 | 74.0 | 78.2 | - |
| Llamba-8B[a] | 69.4 | 61.0 | 54.6 | 82.5 | 77.6 | 80.9 | 73.3 |
| 1st teacher: Llama3.1-8B-Instruct | 73.0 | 68.0 | 55.1 | 81.7 | 79.3 | 81.1 | 73.9 |
| 2nd teacher: Llama3.1-70B-Instruct | 75.7 | 82.3 | 62.4 | 86.7 | 84.7 | 83.7 | 78.7 |
| LoLCATsHedgehog-Llama3.0-8B[b] | - | 23.8 | 40.1 | 72.6 | 65.6 | 76.5 | 53.3 |
| teacher: Llama3.0-8B | 66.6 | 53.3 | 80.1 | 79.1 | 79.9 | 73.1 | 38.1 |
| ARWKV-7B | 62.2 | 62.4 | 52.2 | 79.7 | 76.8 | 79.2 | 68.7 |
| QLinAtt-7B-Instruct | 63.1 | 54.9 | 53.4 | 79.6 | 75.3 | 78.8 | 68.9 |
| QRWKV6-7B-Instruct | 67.5 | 65.7 | 56.3 | 81.4 | 79.0 | 80.3 | 71.1 |
| QRWKV6-7B-Instruct-RoPE | 67.4 | 66.1 | 56.7 | 81.7 | 78.9 | 79.9 | 70.6 |
| QRWKV7-7B-Instruct-RoPE | 68.4 | 68.2 | 55.8 | 81.7 | 79.3 | 79.8 | 71.8 |
| teacher: Qwen2.5-7B-Instruct | 69.6 | 71.7 | 55.0 | 81.5 | 80.5 | 79.4 | 71.4 |
| QRWKV7-7B-Instruct-from72B[a] | 70.7 | 66.7 | 57.2 | 83.3 | 78.2 | 80.1 | 73.9 |
| teacher: Qwen2.5-72B-Instruct | 75.1 | 83.4 | 63.2 | 85.9 | 87.4 | 83.6 | 76.3 |
| QRWKV6-32B-Instruct | 74.2 | 76.6 | 60.9 | 84.3 | 83.0 | 81.2 | 78.2 |
| teacher: Qwen2.5-32B-Instruct | 75.2 | 81.8 | 58.7 | 82.2 | 85.2 | 81.0 | 72.9 |
| **QRWKV6-72B-Instruct** | **75.4** | **77.5** | **63.8** | **86.5** | **85.7** | **82.5** | **79.6** |
| teacher: Qwen2.5-72B-Instruct | 75.1 | 83.4 | 63.2 | 85.9 | 87.4 | 83.6 | 76.3 |

Table 5: Benchmark accuracy scores (%) for student RNN models & transformer teacher models

---

All 0-shot except for 5-shot LolCATs mmlu. Shown are lambada_openai (Paperno et al., 2016), mmlu (Hendrycks et al., 2021a), arc_c_norm, arc_e (Clark et al., 2018), piqa (Bisk et al., 2020), winogrande (Sakaguchi et al., 2021), hellaswag_norm (Zellers et al., 2019).

[a]Not Available: Neither the model weights nor the missing scores were published by the model authors.

[b]Llamba-8B and QRWKV7-7B-Instruct-from72B are distilled from larger 70B and 72B models, respectively, but are compared with Llama3-8B and Qwen2.5-7B-Instruct in Table 5.

[c]The DiJiang paper incorrectly lists 40.3 and 56.1 for arc_c and arc_e respectively. We checked against Mercat et al. (2024)'s evals of this same teacher model and re-ran them here.

| Name | Attention | mmlu↑ | arc_c↑ | arc_e↑ | hella↑ | piqa↑ | winog↑ |
|---|---|---|---|---|---|---|---|
| LoLCATs-Mistralv0.1-7B | SWA 100% | 70.6 | 1.038 | 1.014 | 99.5 | 98.1 | 100.0 |
| LoLCATs-Llama3.1-8B | SWA 100% | 72.7 | 103.7 | 101.5 | 100.1 | 106.9 | 83.9 |
| LoLCATs-Llama3.1-70B | SWA 100% | 79.4 | 99.7 | 96.3 | 99.4 | 97.0 | 80.1 |
| Mamba2-Llama3.0-8B-Instruct | 12.5% | 66.3 | 79.7 | 89.8 | 94.8 | 93.7 | 61.1 |
| Mamba2-Llama3.0-8B-Instruct | 25% | 73.8 | 86.7 | 90.7 | 103.7 | 100.4 | 67.6 |
| Mamba2-Llama3.0-8B-Instruct | 50% | 78.9 | 104.7 | 95.1 | 107.2 | 109.9 | 98.2 |

Table 6: Relative scores for select student hybrid models (still containing some softmax attention) converted from transformer teachers
Relative score is computed as: $\frac{s-r}{t-r}$ where $s$ is student accuracy, $t$ is teacher accuracy, and $r$ is chance of a correct random guess.

| Name | Attention | mmlu↑ | arc_c↑ | arc_e↑ | hella↑ | piqa↑ | winog↑ |
|---|---|---|---|---|---|---|---|
| LoLCATs-Mistralv0.1-7B | SWA 100% | 51.4 | 54.9 | 81.7 | 80.7 | 81.5 | 74.0 |
| teacher: Mistral-7B-v0.1 | 100% | 62.4 | 53.8 | 80.9 | 81.0 | 82.1 | 74.0 |
| LoLCATs-Llama3.1-8B | SWA 100% | 54.9 | 54.4 | 82.4 | 79.1 | 81.0 | 69.7 |
| teacher: Llama-3.1-8B | 100% | 66.1 | 53.4 | 81.5 | 79.0 | 79.0 | 73.5 |
| LoLCATs-Llama3.1-70B | SWA 100% | 67.7 | 60.5 | 85.0 | 84.6 | 821 | .37 |
| teacher: Llama-3.1-70B | 100% | 78.8 | 60.6 | 87.3 | 85.0 | 83.1 | 79.6 |
| Mamba2-Llama3.0-8B-Instruct | 12.5% | 50.8 | 50.3 | 75.8 | 73.1 | 76.8 | 63.4 |
| Mamba2-Llama3.0-8B-Instruct | 25% | 53.7 | 52.5 | 76.4 | 77.7 | 78.7 | 64.8 |
| Mamba2-Llama3.0-8B-Instruct | 50% | 55.7 | 58.2 | 78.8 | 79.5 | 81.5 | 71.5 |
| teacher: Llama3.0-8B-Instruct | 100% | 63.9 | 56.7 | 81.6 | 75.8 | 78.6 | 71.9 |

Table 7: Benchmark accuracy scores (%) for select hybrid models (still containing some softmax attention) converted from attention, and their teacher models

Shown are mmlu (Hendrycks et al., 2021a), arc_c_norm, arc_e (Clark et al., 2018), piqa (Bisk et al., 2020), winogrande (Sakaguchi et al., 2021), and hellaswag_norm (Zellers et al., 2019). All 0-shot except for 5-shot mmlu on LoLCATs and its teachers. Lambda scores were not available for these models.

# 7 Ablation studies

We performed several ablation studies, adding or removing individual mechanisms to the RAD-RWKV6 architecture at the 7B scale. For the "use groupnorm" ablation we added the use of GroupNorm and removed the $k = k * (1 - w)$ state balancing mechanism originally from RWKV6-C2. Results are shown in Table 8.

| Arch | ablation | lmbda↑ | mmlu↑ | arc_c↑ | arc_e↑ | hella↑ | piqa↑ | winog↑ |
|---|---|---|---|---|---|---|---|---|
| RAD-RWKV6 | none | **67.48** | 65.72 | 56.31 | 81.36 | **79.01** | **80.25** | **71.11** |
| RAD-RWKV6 | use rope | **67.40** | **66.10** | **56.66** | 81.65 | **78.86** | 79.92 | 70.56 |
| RAD-RWKV6 | no tokenshift | 67.07 | 65.84 | 55.46 | 80.43 | 78.74 | **80.36** | 68.75 |
| RAD-RWKV6 | no gate | 65.90 | 64.17 | 54.44 | 80.77 | 78.42 | 79.71 | 69.30 |
| RAD-RWKV6 | use groupnorm | 65.59 | 63.40 | **56.48** | **81.86** | 78.65 | 79.05 | 70.32 |

Table 8: Ablation benchmark accuracy scores
All results are Qwen2.5-7B-Instruct converted to RAD-RWKV6 variations after 100M tokens of step 1 training followed by 500M tokens of step 2 training. Arc_c and hellaswag are normalized scores. Teacher model was Qwen2.5B-7B-Instruct except in non-instruct ablation

# 8 What did not work

In the spirit of scientific transparency and to facilitate reproducibility and further experimentation in the field, we share some negative results from our experiments:

**Initial attention matrix orientation**    We originally included an initial "step 0," which was similar to step 1 but used a loss based on attention scores differences. For linear RNNs of the form

$$S_{t+1} = S_t G_t + v_t k_t^T$$

attention scores can be represented as cumulative compositions of $G$ transforms:

$$A_{ij} = q_{ic} (\bigodot_{t=j+1}^{i} G_t)_{cd} k_{jd}$$

This generalizes to Mamba2 (Dao & Gu, 2024), RWKV5, RWKV6 (Peng et al., 2024), RWKV7 (Peng et al., 2025), DeltaNet (Schlag et al., 2021), and Gated DeltaNet (Yang et al., 2025), as well as others. See Table 1 in Peng et al. (2025) for a detailed list of such RNNs.

We hypothesized that matching attention scores up-front would accelerate training compared to Step 1. While final downstream performance was similar, we observed neither benefits in total training time nor lower final loss. On the other hand, extended training of step 0 incurred final higher loss.

**GroupNorm**    Many recent RNN models rely on GroupNorm or LayerNorm to normalize the heads after the attention operation instead of using a denominator as in softmax attention or the classic linear attention formula. We found that for models around 14B and higher, this began to cause serious training instabilities. After examining gradients we theorized that this is because late normalization allows unbounded growth of the inputs to that normalization, which can become catastrophic for deep models. As a result, we applied pre-scaling to keys as in RWKV-6c and a new extension that we developed for RWKV-7, allowing us to remove the normalization and denominator.

**Skipping step 1**    Starting directly with step 2 distillation and skipping step 1 entirely resulted in a much lower performance model. The same level of convergence simply did not occur, with loss plateauing at a higher minimum, even with longer training.

**De-novo initialization of attention weights**    Initializing sequence mixer QKVO weights as if they were untrained, instead of copying them from the teacher model, resulted in consistently worse yet surprisingly reasonable performance. This is compatible with the previous hypothesis stating that factual knowledge is stored mainly in MLP model weights.

**Freezing model weights**    Although it might seem intuitive that the MLPs and embeddings could be frozen during step 2, we find that this results in significantly reduced model performance. We theorize that the internal embedding and hidden state representations need to adapt somewhat to the new RNN sequence mixer's use of channels. Through continued training, the MLPs may learn to route this information differently to avoid conflicts with pre-existing information flow from the teacher model.

**Weight tying**    We initially had trouble obtaining good results when distilling smaller models featuring tied embedding weights. Eventually we found a workaround, which was to allow the embeddings and language modeling head to be untied during distillation, but then discard the updated head at the end and retie the weights. We are uncertain why this worked or whether it will act consistently across architectures. In order to avoid this confounding factor we distilled only models containing untied weights for this paper.

**Larger batchsizes**    The number of optimizer steps appears to be of key importance during conversion. Consequently, increasing batchsize did not help the model converge faster. Additionally, we are careful to keep the learning rate low during steps 2 and 3 in order to avoid excess disturbance to the MLPs. Therefore, adjusting the learning rate higher to compensate for larger batchsizes would be undesirable.

**LoRA training**   We tried using LoRA to perform PEFT, but rank reduction was generally quite detrimental to performance. The one place we found it could work without causing problems was on the embeddings.

**Switching datasets**   We tried using various custom datasets during the context-length extension step. These seemed to create a confused model that would use more and more adjectives as generation progressed. We leave exploration of the ideal dataset for this step to future work.

## 9   Distillation and student architectures

We theorize that mechanisms not present in the distilled teacher model (e.g. token-shift/convolution) offer limited benefit to a distilled student. With longer training these mechanisms could become useful for the converted model. However, our extremely short distillation process probably prevents enough optimization from occurring for the model to learn to put them to good use. Conversely, mechanisms from the teacher (e.g. RoPE) can be important not because they are strictly necessary, but because there are not enough distillation steps for the model to learn to replace them fully. Similarly, we theorize that GQA is probably not an overall benefit in terms of downstream results with enough training, but because the Qwen teacher employed it, it provides a good starting point and guidance for the model during distillation.

## 10   Conclusions

We have shown that RADLADS provides a cost-effective way to convert transformers employing quadratic softmax attention into inexpensive RNN models that feature linear time complexity and constant memory usage. This has potential to save energy, reduce research costs, and enable the rapid testing and deployment of new varieties of RNN architectures. We have already used it to produce and distribute new state-of-the-art open weight linear attention models.

However, RADLADS does have known limitations. Currently, each new architecture design requires meticulous testing to improve its compatibility with the RADLADS protocol. As one example, the GroupNorm or LayerNorm often used in place of a denominator in linear attention variants caused severe training instability at 14B+ scale during distillation and needed replacement. New varieties of model interactions will test these conversions in ways that we cannot predict.

We have many ideas for future work to enhance and extend the RADLADS process. Converting to and from different architectures and testing of more varieties of datasets against each of these would help discover the principles for optimal conversion dataset design. A future direction could be to have the teacher model generate synthetic data, to ensure that the dataset used for conversion remains in-distribution, especially for reasoning models. Another interesting direction is conversion to hybrid models like Goldstein et al. (2024), including conversion that takes into account sparsity or methods of KV Cache compression.

*Acknowledgments*

We would like to thank TensorWave and AMD for sponsoring the initial compute used for this project to train models on MI300X hardware. Eric Alcaide acknowledges support for this work from the Swiss State Secretariat for Education, Research and Innovation (SERI) under contract number 23.00421.

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

# A  Architecture detail

**Notation**  In this section, we use $D$ to denote the model dimension. Bold capital letters represent trainable matrices, and vectors without a subscript $t$ are trainable parameters. The first subscript denotes sequence position and second subscript denotes layer index, where necessary. We use the convention that all vectors are row vectors unless explicitly transposed, so all matrices operate on the right side, therefore $a^T b$ is an outer product and $a b^T$ is an inner one. We use the square subscript to denote a placeholder for variable names and use the $\prod$ sign for cumulative matrix multiplication.

## A.1  RAD-RWKV6 sequence mixing

RAD-RWKV6 is a customized variation of RWKV6-C2 ("Finch-C2") (Goldstein et al., 2024), featuring a Gated Linear Attention (Yang et al., 2024) kernel (so no bonus), and a sigmoid gate. The use of the state balancing technique from RWKV6-C2 enables us to remove state normalization, improving training stability and downstream performance. Removal of bonus did not harm downstream performance. We do not typically use RoPE in this formulation.

The **d**ata-**d**ependent **l**inear int**erp**olation (ddlerp) between $x_t$ and $x_{t-1}$ used in Finch Token Shift is defined as:

$$\text{lora}_\square(x) = \lambda_\square + \tanh(x \boldsymbol{A}_\square) \boldsymbol{B}_\square \tag{1}$$

$$\text{ddlerp}_\square(a, b) = a + (b - a) \odot \text{lora}_\square(a + (b - a) \odot \mu_x) \tag{2}$$

where $\mu_x$ and each $\lambda_\square$ introduce a trainable vector and $\boldsymbol{A}_\square \in R^{D \times z}$ and each $\boldsymbol{B}_\square \in R^{z \times D}$ introduce a pair of low rank trainable matrices, where $z$ is a chosen size for each such pair.

$$r_t = \text{RoPE}(\text{ddlerp}_r(x_t, x_{t-1}) \boldsymbol{W}_r)(d_{head})^{-1/2}, \qquad \text{receptance} \tag{3}$$

$$v_t = \text{ddlerp}_v(x_t, x_{t-1}) \boldsymbol{W}_v, \qquad \text{value} \tag{4}$$

$$g_t = \sigma\left(\text{ddlerp}_g(x_t, x_{t-1}) \boldsymbol{W}_g\right), \qquad \text{gate} \tag{5}$$

$$\tilde{w}_t = \text{ddlerp}_{\tilde{w}}(x_t, x_{t-1}) \boldsymbol{W}_{\tilde{w}}, \qquad \text{decay precursor} \tag{6}$$

$$w_t = \exp(\max(-\exp(\text{lora}_w(\tilde{w}_t)), -5))\}, \qquad \text{decay} \tag{7}$$

$$\tilde{k}_t = \text{ddlerp}_{\tilde{k}}(x_t, x_{t-1}) \boldsymbol{W}_{\tilde{k}}, \qquad \text{key precursor} \tag{8}$$

$$k_t = \tilde{k}_t(1 - w_t) d_k^{-0.5}, \qquad \text{key} \tag{9}$$

Sequence mixing is performed by the $\boldsymbol{wkv}_t$ attention calculation:

$$\boldsymbol{wkv}_t = \sum_{i=1}^{t} \text{diag}\left(\prod_{j=i}^{t-1} w_j\right) k_i^{\text{T}} v_i \in \mathbb{R}^{(D/h) \times (D/h)} \tag{10}$$

The $\boldsymbol{wkv}_t$ attention calculation can alternatively be written in a recurrent manner:

$$\boldsymbol{wkv}_0 = \boldsymbol{0}, \tag{11}$$

$$\boldsymbol{wkv}_t = \text{diag}(w_t) \cdot \boldsymbol{wkv}_{t-1} + k_t^{\text{T}} \cdot v_t \tag{12}$$

$$p_t = r_t \boldsymbol{wkv}_t \tag{13}$$

Finally, the heads are recombined via reshaping so that $p_t \in \mathbb{R}^D$, gated, and transformed into the output as follows:

$$o_t = \left(g_t \odot p_t\right) \boldsymbol{W}_o \in \mathbb{R}^D \tag{14}$$

## A.2 RAD-RWKV7 sequence mixing

RAD-RWKV7 is a customized variation of RWKV-7 (Peng et al., 2025), with tokenshift removed, RoPE applied, and no bonus. The removal of tokenshift speeds up training and inference, and bonus had no beneficial impact on downstream performance. The use of a state balancing technique from RWKV6-C2, modified to apply properly to RWKV-7, enables us to remove state normalization, improving training stability.

$$\text{lerp}(a, b, x) = a + (b - a) \odot x, \tag{15}$$

$$\text{loramlp}_\square(f, x, \text{bias}) = f(x\boldsymbol{A}_\square)\boldsymbol{B}_\square + (\lambda_\square \text{ if bias else } 0), \tag{16}$$

$$a_t = \text{sigmoid}(\text{loramlp}_a(\text{Identity}, x_t, \text{bias=True})), \qquad \triangleright\text{in-context learning rate} \tag{17}$$

$$k_t = \text{RoPE}(x_t\boldsymbol{W}_k), \qquad \triangleright\text{key precursor} \tag{18}$$

$$d_t = \text{loramlp}_d(\text{tanh}, x_t, \text{bias=True}), \qquad \text{decay precursor} \tag{19}$$

$$w_t = \exp(-e^{-0.5}\sigma(d_t)), \qquad \triangleright\text{decay} \tag{20}$$

$$\tilde{k}_t = k_t \odot (1 - w_t + a_t), \qquad \triangleright\text{replacement key} \tag{21}$$

$$v_t = \text{sigmoid}(\text{loramlp}_v(\text{Identity}, x_t, \text{bias=True})), \qquad \text{value residual gate} \tag{22}$$

$$v'_{t,l} = x_t\boldsymbol{W}_v, \qquad \text{value precursor} \tag{23}$$

$$v_t = \begin{cases} v'_{t,0}, & \text{layer } l = 0 \\ \text{lerp}(v'_{t,0}, v'_{t,l}, v_t), & \text{layer } l \geq 1 \end{cases}, \qquad \triangleright\text{value} \tag{24}$$

$$r_t = \text{RoPE}(x_t\boldsymbol{W}_r)(d_{head})^{-1/2}, \qquad \triangleright\text{receptance} \tag{25}$$

$$g_t = \text{loramlp}_g(\sigma, x_t, \text{bias=False}) \qquad \triangleright\text{rwkv gate} \tag{26}$$

After weight preparation, we reshape $(r, w, \tilde{k}, v, \kappa, a)_t$, splitting them to $h$ heads, with each head sized $D/h$. We always assume that $h$ is a factor of $D$ and heads are equally split. All operations in this section are shown per-head.

Before mixing in the time dimension, $k_t$ is normalized per head to form the removal key $\kappa_t$:

$$\kappa_t = k_t/\|k_t\|_2 \qquad \triangleright\text{normalized removal key} \tag{27}$$

Sequence mixing is performed by the $\boldsymbol{wkv}_t$ attention calculation:

$$\boldsymbol{wkv}_t = \sum_{i=1}^{t}\left(v_i^T\tilde{k}_i \prod_{j=i+1}^{t}\left(\text{diag}(w_j) - \kappa_j^T(a_j \odot \kappa_j)\right)\right) \in \mathbb{R}^{(D/h)\times(D/h)} \tag{28}$$

The $\boldsymbol{wkv}_t$ attention calculation can alternatively be written in a recurrent manner:

$$\boldsymbol{wkv}_0 = \boldsymbol{0}, \tag{29}$$

$$\boldsymbol{wkv}_t = \boldsymbol{wkv}_{t-1}\left(\text{diag}(w_t) - \hat{\kappa}_t^T(a_t \odot \hat{\kappa}_t)\right) + v_t^T \cdot \tilde{k}_t \tag{30}$$

$$p_t = r_t\boldsymbol{wkv}_t^T \qquad \triangleright\text{attention result} \tag{31}$$

Finally, the heads are recombined via reshaping so that $p_t \in \mathbb{R}^D$, gated, and transformed into the output as follows:

$$o_t = (g_t \odot p_t)\boldsymbol{W}_o \in \mathbb{R}^D \tag{32}$$

### A.3 Linear Attention sequence mixing

The Linear Attention used to train the QLinAtt-7B-Instruct model in the results section is a slight variation of traditional linear attention. It uses no $\phi$ function, and applies LayerNorm per head to normalize the attention results, formally stated per head as:

$$p_t = \text{LayerNorm}(\text{RoPE}(x_t W_q)\text{RoPE}(x_t W_k)^T M(x_t W_v)) \tag{33}$$

Finally, the heads are recombined via reshaping so that $p_t \in \mathbb{R}^D$, and transformed into the output as follows:

$$o_t = p_t W_o \in \mathbb{R}^D \tag{34}$$

## B Additional training settings

We manually choose between DeepSpeed (Li et al., 2024a) ZeRO stages 1 or 2 and FSDP (Zhao et al., 2023), depending on VRAM availability. Typically we use DeepSpeed ZeRO stage 1 for our step 1 whenever possible, as it results in faster training and step 1 requires less VRAM than step 2. We find that FSDP is more VRAM efficient than DeepSpeed ZeRO Stage 3 for larger models, and we generally use it for steps 2 and 3.

## C Absolute scores summary

| Name | Student / Teacher accuracy (%) | | |
| --- | --- | --- | --- |
| | lmbda↑ | mmlu↑ | Others↑ |
| SUPRA | 69.2 / 75.8 | 33.1 / 62.4 | 69.8 / 74.4 |
| LOLCats / Hedgehog[a] | - | 23.8 / 66.6 | 61.6 / 73.1 |
| MOHAWK | 50.1 / 53.4 | 24.2 / 41.8 | 65.1 / 67.2 |
| Mamba in the Llama | 43.9 / 72.2 | 45.2 / 63.9 | 65.4 / 72.9 |
| DiJiang[a] | - | 40.7 / 46.8 | - |
| ARWKV | 62.2 / 69.6 | 62.4 / 71.7 | 71.3 / 73.5 |
| Llamba | 69.4 / 73.0 | 61.0 / 68.0 | 73.8 / 74.2 |
| **RADLADS (ours)** | 68.4 / 69.6 | 68.2 / 71.7 | 73.7 / 73.5 |

Table 9: Absolute accuracy scores of student versus teacher model for recent softmax attention to purely recurrent model conversions up to 8B parameters

[a] Neither model weights nor the missing scores were published for this pure RNN model by its authors.

## D Advanced and long-context benchmarks

After using the RADLADS protocol to create the RNN models featured in this paper, we became interested in trying the technique to adapt teacher models to non-RNN forms. To this end, we distilled Qwen3-8B-Base to utilize a new attention variant named Softpick attention(Zuhri et al., 2025). The results for select advanced benchmarks are shown in Tables 10 and 12.

Performance was excellent, rivaling or exceeding the original teacher model on every test. This was encouraging, since Softpick attention is quite different from normal attention, and removes all negative attention scores. However, the distillation failed to achieve the primary goal of Softpick attention, which is to avoid the creation of attention sinks. Analyzing the attention maps showed that our distilled model still contained these sinks. Despite not achieving the goals of the Softpick architecture in this case, we believe that the ability to inexpensively convert to new non-RNN forms of attention may be a useful tool for researchers.

| model | gsm8k(%) | HumanEval(%) | passkey>=95 (ktok) |
|---|---|---|---|
| QRWKV7-Qwen2.5-7B | 61.4 / 85.9 | 50.0 / 79.9 | 8k / 34k+ |
| ARWKV | 53.4 / 85.9 | 50.0 / 79.9 | 1k / 34k+ |
| Llamba-8B | 56.0 / 81.0 | 6.1 / 56.1 | 1k / 128k* (*Meta's claim) |
| Softpick-Qwen3-8B-Base | 85.7 / 86.1 | 81.7 / 80.5 | 34k+ / 34k+ |

Table 10: Reasoning and long context benchmark scores (student / teacher)
These QRWKV and Softpick models were distilled using only steps 1 and 2a
GSM8K benchmark was run with max_new_tokens=512

| model | ctx | RULER score |
|---|---|---|
| QRWKV7-7B-Instruct | 4096 | 39.7 |
| QRWKV7-7B-Instruct | 8192 | 26.1 |
| QRWKV7-7B-Instruct | 16384 | 16.5 |
| teacher: Qwen2.5-7B-Instruct | 16384 | 69.7 |

Table 11: RULER(Hsieh et al., 2024) benchmark scores (%)

| Model | Max allowed response length | Accuracy |
|---|---|---|
| QRWKV7-Qwen3-8B | 1024 | 25.9 |
| teacher: Qwen3-8B | 1024 | 41.4 |
| QRWKV7-Qwen3-8B | 2048 | 27.6 |
| teacher: Qwen3-8B | 2048 | 59.9 |
| Qwen3-Softpick-8B-Base | 1024 | 53.0 |
| teacher: Qwen3-8B-Base | 1024 | 53.1 |

Table 12: Benchmark accuracy scores (%) for Minerva Math(Hendrycks et al., 2021b; Lewkowycz et al., 2022) reasoning test

## E   Inference speed

In order to demonstrate the improvement in inference speed achievable by converting from Qwen to QRWKV7 via the RADLADS process, we test various context lengths using an optimized vLLM instance on 1x H100 in bfloat16, with 32 simultaneous requests. We find that with as few as 8k tokens in and 256 tokens out, there is a significant speed increase. This increase continues to widen as the number of output tokens increases. RAD-RWKV embeddings and MLP layers are the same shapes and use the same computation as the teacher model. Therefore, all speed improvement comes from the attention replacement, which uses a fixed size state and much lower memory bandwidth.

| Tokens in | Tokens out | QRWKV7-32B | Qwen3-32B | ratio |
|---|---|---|---|---|
| 8k | 256 | 86.5 sec | 139.5 sec | 1.61x |
| 7k | 1k | 148.7 sec | 411.7 sec | 2.77x |
| 6k | 2k | 229.6 sec | 783.0 sec | 3.41x |

Table 13: Inference speed benchmark of converted versus teacher model

