# OpenReview forum: "RADLADS: Rapid Attention Distillation to Linear Attention Decoders at Scale"
_colmweb.org/COLM/2025/Conference — COLM 2025_

### Official Review · Reviewer_1f3Y · 2025-05-07

**Rating:** 7
**Confidence:** 5
**Ethics Flag:** 1

**Summary:**

This work is about distilling pretrained Transformers into RNNs without sacrificing much performance. The proposed method follows a three-stage strategy:

Stage 1: Hidden State Alignment – Each layer of the student RNN is trained to match the hidden states of the corresponding layer in the teacher Transformer. The authors adopt a simple end-to-end training approach and show the effectiveness.

Stage 2: Logit Distillation – The student is further trained to match the teacher's output logits, a technique used in prior literature.

Stage 3: Full-parameter Fine-tuning on longer context. The authors find that LoRA-based finetuning is less suitable in this Transformer-to-RNN distillation setting.

Overall, the paper explores several known distillation stages and simplifies the pipeline by retaining only the components that empirically contribute to performance. A key contribution is the use of more expressive RNN backbones—specifically, the RAD-RWKV7 variant—which improves alignment and downstream performance. This is a notable departure from prior work that largely relied on vanilla or gated version of linear attention (e.g., GLA, Mamba2). The work is timely and well-aligned with the recent rapid development in scalable RNNs.

**Questions To Authors:**

- Figure 2 (third row): It seems MOHAWK also includes an attention hidden-state alignment stage, yet this is not reflected in the figure. Could the authors clarify or update the figure accordingly?

- Line 222: Several referenced models (e.g., Mamba2, DeltaNet, Gated DeltaNet) are mentioned without proper citations. Please include appropriate references.

- Section 7: While it is helpful to see which design choices did not work, it would be more informative if quantitative metrics were provided to illustrate the extent of degradation for each failed approach.

**Reasons To Accept:**

- The paper is very honest about what works and what doesn’t, providing detailed negative results and practical insights

- Achieves strong empirical performance with only 350–700M training tokens, significantly less than prior work

- Demonstrates the importance of using more advanced RNN variants (RAD-RWKV6/7), which yield substantial gains over earlier architectures

**Reasons To Reject:**

I am open to increasing my score if the authors can address my concerns:

- (Important) Regarding evaluation, the current evaluation only covers a limited set of standard benchmarks. I would like to see performance on more challenging reasoning tasks (e.g., GSM8K, HumanEval) and long-context benchmarks (e.g., RULER, LongBench), where RNNs often underperform compared to full attention models.

- (Somewhat minor) The paper focuses exclusively on the RWKV family. It would strengthen the work to include comparisons or ablations involving alternative recurrent architectures such as DeltaNet, Gated DeltaNet, TTT, Titans, or DeltaProduct.

- (Somewhat minor)  Regarding presentation, the main reported numbers are relative gains over random guessing. While I understand the rationale, I find it less intuitive than reporting absolute accuracy values, which would aid comparison.

---

> ### Author Response · Authors · 2025-06-03
>
> Thank you for your review and detailed set of concerns. We have done our best to address them and offer a set of proposed changes below that we believe will significantly improve the paper:
>
> > …more challenging reasoning tasks (e.g., GSM8K, HumanEval) and long-context benchmarks (e.g., RULER, LongBench)...
>
> Thank you for making this important point. We were able to obtain the following results for certain 7-8B models, shown in the format student/teacher score:
>
> | test | RADRWKV7 | ARWKV | Llamba-8B |
> |-|-|-|-|
> |gsm8k|61.4%/85.9%|53.4/85.9%|56.0%/81.0%|
> |HumanEval|50.0%/79.9%|50.0%/79.9%|6.1%/56.1%|
> |passkey >= 95%|8k/34k+|1k/34k+|1k/128k* (*Meta's claim)|
>
>
> Unfortunately, some authors did not release model weights, and some models emit errors when running ‘generate-until’ style benchmarks, and/or require extensive code changes. We will continue our attempts to obtain these benchmarks for the other models for the camera-ready version.
>
> In the RADRWKV7 evals above, we made some small changes to our training hyperparameters that we will add for the camera-ready version, which improve context length usage and can replace stage 3. Specifically, we used a sequence length of 4096 with a reduced total batch size of 32 during stage 2, alongside a new secondary 1e-4 learning rate applied to the RWKV layers only.
>
> > ...paper focuses exclusively on the RWKV family...
>
> We found that RWKV-6 did not work well for conversion. Although RAD-RWKV6 has a name that includes “RWKV”, it is actually more similar to Gated Linear Attention (GLA), and uses the GLA kernel paired with RWKV-6c’s method of maintaining a state with limited numerical size. This choice was motivated by a hunch that the ‘bonus’ term from RWKV-6 was problematic, which is the main difference between the RWKV-6 and GLA kernels. This paper demonstrated two different architectures, based on GLA and RWKV-7 respectively. DeltaNet and Gated DeltaNet are mainly subsets of RWKV-7, and ablation studies on these aspects are covered in the cited RWKV-7 paper. We have also used the RADLADS process to convert Qwen2.5-7B-Instruct to the new ‘softpick attention’ featured in a recent preprint https://arxiv.org/abs/2504.20966, demonstrating its utility for research even on non-linear architectures.
>
> | arch | lambada ppl | lambada acc | mmlu | arc_c_n | arc_e | hella_n | piqa | winog |
> |-|-|-|-|-|-|-|-|-|
> |softpick|3.67/3.66|71.6%/71.0%|74.6%/74.7%|57.5%/56.3%|82.1%/81.9%|78.5%/78.6%|79.2%/79.3%|73.5%/73.0%|
>
> > …less intuitive than reporting absolute accuracy values…
>
> Thank you for pointing this out. We struggled with this, and our early drafts showed only absolute accuracy values. Though imperfect, the relative scores communicate two important components: the amount learned from the teacher, as well as the amount learned relative to random guessing. Without these it was unclear that some models learned nothing on certain topics. We acknowledge that the terminology needs improvement, and have made changes to the manuscript to ensure that the reader is made aware of the differences between absolute and relative scores. We will also make sure that the camera ready paper shows results for both methods in every section, including the first table.
>
> > Figure 2 (third row): It seems MOHAWK also includes an attention hidden-state alignment stage, yet this is not reflected in the figure. Could the authors clarify or update the figure accordingly?
>
> Thank you, this was an oversight that unfortunately did not get corrected until after submission. MOHAWK also does not use attention weights transfer. These errata have now been corrected in the manuscript. The correct sequence now listed for MOHAWK/Llamba is Attention Matrix Orientation -> Attention Hidden-State Alignment -> Knowledge Distillation
>
> > Line 222: Several referenced models (e.g., Mamba2, DeltaNet, Gated DeltaNet) are mentioned without proper citations. Please include appropriate references.
>
> Thank you for this important note. We have added these and will double check to make sure we have all other citations.
>
> > Section 7: While it is helpful to see which design choices did not work, it would be more informative if quantitative metrics were provided to illustrate the extent of degradation for each failed approach.
>
> We have the following RAD-RWKV6 architecture ablations available to add to the paper:
>
> | ablation | lambada↑ | MMLU | arc_c | arc_e | hella | piqa  | winog |
> |-|-|-|-|-|-|-|-|
> |none|0.6748|0.6572|0.5631|0.8136|0.7901|0.8025|0.7111|
> |use rope|0.6740|0.6610|0.5666|0.8165|0.7886|0.7992|0.7056|
> |no tokenshift|0.6707|0.6584|0.5546|0.8043|0.7874|0.8036|0.6875|
> |no gate|0.6590|0.6417|0.5444|0.8077|0.7842|0.7971|0.6930|
> |use groupnorm|0.6559|0.6340|0.5648|0.8186|0.7865|0.7905|0.7032|
>
> If it would be helpful to include in the paper, we would be happy to run and provide some additional ablation experiments for the ‘what did not work’ section, such as skipping step 1, freezing model weights, and using larger batch sizes.

---

> > ### Author Response · Authors · 2025-06-06
> >
> > In order to further demonstrate the applicability of the method to other linear attention variations, we have distilled a simple linear attention conversion from Qwen2.5-7B-Instruct, with groupnorm replacing the need for a positive-only phi function. Please find benchmark results below:
> >
> > | arch | lambada ppl | lambada acc | mmlu | arc_c_n | arc_e | hella_n | piqa | winog |
> > |-|-|-|-|-|-|-|-|-|
> > |linatt groupnorm|4.93|63.1%|54.9%|53.4%|79.6%|75.3%|78.8%|68.9%|
> > |Qwen2.5-7B-Instruct (teacher)|3.69|69.6%|71.7%|55.0%|81.5%|80.5%|79.4%|71.0%|

---

> > ### Comment · Reviewer_1f3Y · 2025-06-07
> >
> > Thank you for sharing the new results. It looks like GSM and HumanEval still suffer from significant performance gaps. Could you highlight these limitations more explicitly in future revisions? I appreciate the honesty in your writing—being transparent about weaker results is equally important.
> >
> > Second, I find the pass-key retrieval task a bit too simple. Would it be possible to evaluate on the entire RULER benchmark and also LongBench instead? I understand that other baselines don’t provide generate_utils, which is fine—but it would still be helpful to compare against the teacher Transformer models if possible. Again I want to understand how large the performance gap between distilled linear models and pretrained Transformers.
> >
> > It’s encouraging to see that your training recipe transfers well to other attention variants like softpick without degrading performance. Do you think the performance gap stems from inherent limitations in recurrent architectures? It might be worth exploring a hybrid architecture in your next iteration.

---

> > > ### Author Response · Authors · 2025-06-07
> > >
> > > Thank you for encouraging us to run these - they have been informative, even to us. And we agree that weaker results are just as important to show so that readers can understand an accurate comparison. We will make sure to highlight these limitations, and indeed any others we are able to find prior to camera-ready, in revisions to the paper.
> > >
> > > It's challenging for us to run the entire RULER and LongBench suites, which is what has prevented us from trying these until now. We will attempt to do so, and will see if we can report back here within the remaining time, at least for selected subtasks if we cannot manage the entire run. One potential issue is that sometimes the optimal manner of prompting can vary between transformers and recurrent models, and this may not be reflected in the tasks. It may indeed show significantly lower performance than Transformers, and we feel that this is important to report in the paper if it does turn out that way.
> > >
> > > Regarding transference of the recipe to other attention variants, we think that this presents a valuable opportunity for researchers to be able to try out other attention mechanisms at scale without the extreme costs associated with pretraining, even if it is not a perfect substitute for full pretraining.
> > >
> > > We are actively exploring hybrid architectures, with the goal being to retain the full context length usage of a transformer, and by extension its abilities on these difficult tasks, while reducing inference costs. As for whether or not the performance gap is inherent to the recurrent architecture, there seems to be some lack of certainty in either direction. We have seen evidence that recurrent models can excel at many tasks, but whether or not they can learn all of these behaviors rapidly via distillation may still be an open question. Hopefully further research can reveal the answer in future papers.

---

> > > > ### Comment · Reviewer_1f3Y · 2025-06-07
> > > >
> > > > Given the promise to provide more results in a future revision, I am increasing my score to 7.

---

> > > > > ### Comment · Reviewer_1f3Y · 2025-06-09
> > > > >
> > > > > Actually can you also report the results of GSM8K/HumanEval/etc.. for the softpick model when you have a chance for the camera ready version? It would be an interesting experimental standard for designing new softmax attention mechanism if the results are really good

---

> > > > > > ### Author Response · Authors · 2025-06-09
> > > > > >
> > > > > > That's a good idea. Right now our code for evals on softpick is pretty limited and slow due to lack of HF and therefore multi-GPU support - but if we can get the full model code from them we should be able to run these.
> > > > > >
> > > > > > Also, we just got some new unexpected feedback that apparently the converted softpick model we made does a sort of an end run around the goals of the softpick project. Softpick was attempting to remove attention sinks as a behavior, partly to make quantization work better. But apparently, in the converted model, when the attention scores are visualized the attention sinks were only slightly diminished (scores no longer sum to 1.0) but did still exist.
> > > > > >
> > > > > > It's very interesting that it was able to perform so well after conversion (unless we end up seeing lower gsm8k/humaneval results), because the calculation is quite different from normal attention. They also now trained a larger 1.8B model (their original was only 340M) and found that this 1.8B did not perform well on benchmarks, whereas we can see that the conversion did! So the conversion to softpick contains several fairly unexpected results. Maybe some new kind of initial step to cut out or discourage the attention sinks would make it possible to see how conversion, and softpick itself, proceeds in that case. That would be getting quite far beyond the scope of the RADLADS paper, which is really just about the conversion process and linear models, but we're intrigued to see how conversion through distillation could open up some new territory for researchers.

---

> > ### Author Response · Authors · 2025-06-10
> >
> > We were able to finish running RULER at a various context lengths:
> >
> > |model|ctx|RULER score|
> > |-|-|-|
> > |RAD-RWKV7 Qwen2.5-7B-Instruct|4096|39.7%|
> > |RAD-RWKV7 Qwen2.5-7B-Instruct|8192|26.1%|
> > |RAD-RWKV7 Qwen2.5-7B-Instruct|16384|16.5%|
> > |Qwen2.5-7B-Instruct|16384|69.7%|
> >
> > Here are the other tests above revised to include the softpick Qwen3-8B-Base conversion as well vs its teacher Qwen3-8B-Base:
> >
> > | test | RADRWKV7 | ARWKV | Llamba-8B | Softpick-Qwen3-8B-Base |
> > |-|-|-|-|-|
> > |gsm8k|61.4%/85.9%|53.4/85.9%|56.0%/81.0%|85.7%/86.1%|
> > |HumanEval|50.0%/79.9%|50.0%/79.9%|6.1%/56.1%|81.7%/80.5%|
> > |passkey >= 95%|8k/34k+|1k/34k+|1k/128k* (*Meta's claim)|34k+/34k+|
> >
> > |model|max allowed response length|score|
> > |-|-|-|
> > |RAD-RWKV7|1024|25.9%|
> > |Qwen3-8B|1024|41.4%|
> > |RAD-RWKV7|2048|27.6%|
> > |Qwen3-8B|2048|59.9%|
> > |Qwen3-Softpick-8B-Base|1024|53.0%|
> > |Qwen3-8B-Base|1024|53.1%|
> >
> > There seems to be effectively no degradation for softpick. Interestingly, the softpick version even exceeded its teacher model on humaneval.

---

> > > ### Comment · Reviewer_1f3Y · 2025-06-10
> > >
> > > Thank you for the new results — they’re very informative! Please make sure to emphasize the limitations of distilling Transformers into RNNs, especially for long-sequence tasks. Common sense reasoning and MMLU are knowledge-intensive and more closely tied to the FFNs, whereas RULER focuses on in-context recall, an area where RNNs tend to fall short. I'm looking forward to the hybrid model results — very curious to see how the trade-offs play out.
> > >
> > > Also, the SoftPick results are quite impressive. I think there’s a strong case for highlighting the training recipe itself as a central contribution of the paper, beyond just the architecture-level distillation. This could be important for future work on distilling Transformers into other architectures, including softmax attention variants.
> > >
> > > Overall, I find this to be a strong submission and recommend it for clear acceptance

---

### Official Review · Reviewer_CHRt · 2025-05-15

**Rating:** 8
**Confidence:** 4
**Ethics Flag:** 1

**Summary:**

The authors introduce a novel protocol for converting large softmax attention transformer models into linear attention models (specifically, RWKV variants) with minimal training data and cost. The process involves a three-step distillation pipeline and the creation of two new architectures (RAD-RWKV6 "RADFinch" and RAD-RWKV7 "RADGoose") to facilitate efficient conversion. The resulting models retain performance close to their original transformer counterparts but benefit from the efficiency of linear attention, especially for long-sequence inference. The paper is very clearly written, includes a nice set of experimental comparisons both to other distillation approaches and to different architecture variants that the authors considered.

**Questions To Authors:**

Given the significantly smaller about of data used by your distillation method, isn't there a stronger dependency between the choice of the data set and the downstream evaluation/usage as compared to methods that require/use more data?

**Reasons To Accept:**

- The distillation methods proposed in this paper will be of significant interest to both researchers and pracitioners due to their low cost (~$2k) and the computational savings that the resulting models allow.
- The paper contains a thorough empirical analysis of the approach and relevant baselines.

**Reasons To Reject:**

- Nothing stands out

---

> ### Author Response · Authors · 2025-06-03
>
> Thank you for the review and the excellent question, which we hope to answer in the camera-ready version of the paper. Please see our response below:
>
> > Given the significantly smaller amount of data used by your distillation method, isn't there a stronger dependency between the choice of the data set and the downstream evaluation/usage as compared to methods that require/use more data?
>
> This was a very interesting question to us as well. We think that training on less data provides fewer opportunities for the data to influence the model, but this is somewhat learning rate dependent as well. Choice of dataset did appear to matter significantly in terms of benchmark results. Unfortunately, we did not have the resources to try a large variety of different datasets. One idea we have for future work is to try using synthetic data generated from the teacher model itself to do the distillation.
>
> After submission we tried converting Qwen3, which is heavily CoT oriented. Conversion was successful using just DCLM as a dataset, and normal benchmarks seemed good. But during longer evals designed to test reasoning it got into strange loops between the `<think>` and `</think>` tokens. Once we mixed in 10% CoT data to the dataset used for distillation this looping behavior went away. We would like to add this finding to the paper for the camera-ready version.

---

> > ### Comment · Reviewer_CHRt · 2025-06-07
> > **Response to authors**
> >
> > Thank you for the additional update. Adding any such empirical results relating the distribution of distillation data to the benchmarks would definitely be welcome in the final version.

---

### Official Review · Reviewer_emZF · 2025-05-16

**Rating:** 7
**Confidence:** 4
**Ethics Flag:** 1

**Summary:**

This paper focuses on distilling softmax-attention transformers models into linear attention decoder models, using a small pretraining token budget and cost. The distillation process has several stages: attention weights transfer, attention hidden-state alignment, knowledge distillation, and finetuning. In attention weights transfer, attention weights from teacher is used to initialize student attention weights. In attention hidden state alignment, linear transformation layers are added for each teacher attention layer and trained parallely. In knowledge distillation stage, KL divergence on student and teacher logits is used. In finetuning, student model is trained on longer sequences. The distilled model achieve similar quality to original transformer, achieve SOTA downstream performance compared to linear attention models.

**Questions To Authors:**

See weaknesses.

**Reasons To Accept:**

- The proposed pipeline requires a lot less (0.005% of teacher training tokens) pretraining token budget compared to other methods, which leads to efficient training.
- The distilled models achieve SOTA performance for linear attention models of their size.
- The introduction of RAD-RWKV6 and RAD-RWKV7 for conversion is an interesting contribution.

**Reasons To Reject:**

- RAD-RWKV7 seems to have training stability issues at larger parameter/layer counts. While acknowledged as ongoing work, this limitation is significant for scaling the approach to even larger models and could impact the reliability of conversions at the extreme end of the scale.
- Lack of evaluation on reasoning benchmarks, given that "the details of how the conversion process impacts reasoning models like our converted QwQ model is still unknown and requires further testing."
- Deeper discussion on why certain components become less relevant during distillation (compared to pretraining) could provide more generalizable insights for future architecture design.
- While the pretraining token budget is smaller, its' not clear how the training time compares to other approaches.
- While the paper mentions O(1) time per token and avoidance of KV cache, more detailed quantitative benchmarks on inference speed (tokens/second) and memory usage (GB) for the converted models across different sequence lengths, compared to their teacher models, would further strengthen the claims of efficiency.

See updates in the comment section.

---

> ### Author Response · Authors · 2025-06-03
>
> Thank you for the detailed review and important topics you’ve raised. Please see our answers and proposed changes below:
>
> > RAD-RWKV7 seems to have training stability issues at larger parameter/layer counts…
>
> This was indeed a serious problem. Fortunately, we have now discovered the reason for the instability and corrected it!
> In deeper models the GroupNorm present at the end of the time mixer appears to allow very large activations and gradients to form prior to it. We developed a variation on the RWKV-6c mechanism for numerical state size limiting that can be applied to RWKV-7. This allowed us to remove the GroupNorm and stabilized training at larger scales. The core idea is to normalize the keys and apply k*=1-w+a instead of k*=1-w found in RWKV-6c and RAD-RWKV6. We have tested this up to 32B scale and will supply this updated formula with detailed rationale and release a trained model for the camera-ready version of the paper.
>
> > Lack of evaluation on reasoning benchmarks…
>
> Reasoning models were not a significant focus of this paper, and were popularized after we had developed the main process, and we quickly converted QwQ just to see if the process still worked on it. We recently also used RADLADS to convert Qwen3 thinking models. Using DCLM we saw repetitive looping behaviors during reasoning, but found these can be eliminated by training on a mixture of 90% DCLM and 10% OpenThoughts, leading to a reasonable chain of thought. We are still working on increasing the performance on long-form reasoning benchmarks. It is possible that reasoning datasets must be much more aligned with the teacher model than the ones we are using. A future direction could be to have the teacher model generate synthetic data, to ensure that the dataset used for conversion remains in-distribution.
>
> > Deeper discussion on why certain components become less relevant during distillation…
>
> Thank you, this is a great point and we will add this for camera-ready. Our general view is that mechanisms not present in the teacher model (e.g. tokenshift/convolution) are ones with limited benefit for distillation. With longer training these mechanisms could become useful for the converted model. But our extremely short distillation process prevents enough optimization from occurring for the model to learn to put them to good use. Conversely, mechanisms from the teacher (e.g. RoPE) are important not because they are strictly necessary, but because there are not enough distillation steps for the model to learn to replace them fully. Similarly, GQA is probably not a benefit with enough training, but because the Qwen teacher employed it, it provides a good starting point and guidance for the model during distillation. We will add more discussion around this topic to the paper for the camera-ready version.
>
> > …not clear how the training time compares to other approaches
>
> Since many of the methods share algorithmically equivalent steps, comparisons can be made on this basis. In our opinion, it is more appropriate for research of this nature to compare algorithms used for each stage than the optimization of the specific models and training pipelines.
>
> For example, Llamba 8B shares two stages with RADLADS: 5B vs 100M tokens of Attention Hidden-State Alignment (AHSA) and 6.5B vs 250-500M tokens of Knowledge Distillation (KD). Llamba adds 500M tokens of Attention Matrix Orientation (AMO), and RADLADS has an optional 100M tokens of Fine-Tuning (FT). The AHSA and KD stages follow the same algorithms in both methods, with the only significant per-token speed differences coming from trainer optimization and attention mechanism speeds. Even if Mamba were 2x as efficient as RWKV, the 50:1 ratio of AHSA tokens and 10:1 ratio of KD would dominate. Furthermore, we do not require AMO, and that phase alone takes nearly as many tokens for Llamba as our entire process.
>
> > more detailed quantitative benchmarks on inference speed (tokens/second) and memory usage (GB)
>
> This is a great suggestion for how we can demonstrate the efficiency claims. In response to your request, we ran an experiment to show the comparative speed improvement using the highly optimized vllm platform to process 32 requests on a single H100 80GB in bfloat16 precision:
>
> | tok in | tok out | RADRWKV7-32B | Qwen3-32B | ratio |
> |-|-|-|-|-|
> |8k|256|86.5 sec|139.5 sec|1.61x|
> |7k|1k|148.7 sec|411.7 sec|2.77x|
> |6k|2k|229.6 sec|783.0 sec|3.41x|
>
> The cited RWKV-7 paper also contains detailed speed and memory comparisons that show the RWKV kernel tends to overtake Flash Attention in speed terms at around 2-4k context length. We can add a chart of these speeds for camera-ready. Our embeddings and FFN layers are the same speed as the teacher model.
>
> We hope that we have adequately addressed any concerns you may have and that on the basis of these proposed changes to the paper you will consider increasing your score. If you have other questions or concerns please do not hesitate to ask.

---

> > ### Author Response · Authors · 2025-06-06
> >
> > In order to help address the concern about lack of evaluation on reasoning benchmarks, we were able to run some evaluations on minerva_math, which is a superset of the MATH500 reasoning benchmark. These were performed on our conversion of Qwen3-8B to RAD-RWKV7, as well as the teacher model. Please find results below:
> >
> > |model|max allowed response length|score|
> > |-|-|-|
> > |RAD-RWKV7|1024|25.9%|
> > |Qwen3-8B|1024|41.4%|
> > |RAD-RWKV7|2048|27.6%|
> > |Qwen3-8B|2048|59.9%|
> >
> > The converted model does not appear to improve as much with additional output tokens as the teacher model.

---

> > > ### Comment · Reviewer_emZF · 2025-06-07
> > >
> > > Thanks for your detailed response.
> > >
> > > * It's nice that training instability has been diagnosed, I believe this finding about GroupNorm and further development of state size limiting mechanism strengthens the paper considerably, I look forward to seeing the updated formula in the revision.
> > >
> > > * I appreciate the new experiments converting Qwen3 thinking models and the discussion of the results. Finding that a data mixture can eliminate looping behavior is a valuable insight, and thoughts on future directions are well-reasoned.
> > >
> > > * Thank you for agreeing to add a more detailed discussion on why certain components become less relevant during distillation. The rationale you provided in the rebuttal is insightful and adds important context to your methodology.
> > >
> > > * The clarification on training time comparison by focusing on algorithmic stages and token counts is a much clearer way to frame the efficiency of RADLADS. Most importantly, the new quantitative inference speed benchmarks on the H100 are impressive.
> > >
> > > Planned revisions and the new results provided by authors have thoroughly addressed my initial concerns. On the basis of this strong rebuttal and the planned improvements, I have raised my score.

---

> ### Comment · Area_Chair_Drgh · 2025-06-07
> **.**
>
> Dear reviewer, could you please respond to the rebuttal?

---

### Decision · Program_Chairs · 2025-07-08

**Decision:**

Accept

**Comment:**

Converting pretrained Transformers into more-efficient linear Transformers/SSMs is an important open problem. This paper presents an effective recipe for doing so, and shows that it is possible to convert the Qwen series of models into RWKV variants. All the reviewers appreciated the effectiveness of the approach, as well as a thorough discussion of negative results (section 7). This is a clear accept.There were slight concerns with regard to whether such models would work on longer context tasks, as well as some minor presentation issues some numbers (e.g., ratios vs. absolute numbers). I encourage the authors to address these for the camera-ready.